# Learning to Abstract Visuomotor Mappings using Meta-Reinforcement Learning

**Carlos A. Velazquez-Vargas** [1] , **Isaac Ray Christian** [1], **Jordan A. Taylor** [1,2], **Sreejan Kumar**[2,3]

1 Princeton University Department of Psychology
2 Princeton Neuroscience Institute
3 NYU Center for Data Science

## Abstract

We investigated the human capacity to acquire multiple visuomotor mappings for *de novo* skills. Using a grid navigation paradigm, we tested whether contextual cues implemented as different "grid worlds", allow participants to learn two distinct key-mappings more efficiently. Our results indicate that when contextual information is provided, task performance is significantly better. The same held true for meta-reinforcement learning agents that differed in whether or not they receive contextual information when performing the task. We evaluated their accuracy in predicting human performance in the task and analyzed their internal representations. The results indicate that contextual cues allow the formation of separate representations in space and time when using different visuomotor mappings, whereas the absence of them favors sharing one representation. While both strategies can allow learning of multiple visuomotor mappings, we showed contextual cues provide a computational advantage in terms of how many mappings can be learned.

## 1 Introduction

There has been considerable interest in determining how contextual cues allow the consolidation and retrieval of multiple visuomotor memories (Howard et al., 2013; Heald et al., 2018; 2023a). While it has been shown that arbitrary external contextual-cues – such as colors, sounds or shapes – are effective in separating the task contingencies in a variety of domains such as in classical conditioning (Gershman, 2017), episodic memory (Pu et al., 2022) and value-based decision making (Bornstein & Norman, 2017), the same cues cannot prevent catastrophic interference in visuomotor adaptation tasks (Howard et al., 2013) unless they are presented in close temporal proximity (Avraham et al., 2022).

In contrast with these laboratory-based findings, we appear capable of storing multiple mappings when using a variety of digital devices despite having similar movements (e.g., video games use the same controller to play a racing game and a first-person shooter). One interpretation of the above, is that a great proportion of these mappings may not be the result of visuomotor adaptation but of instead of *de novo* skill learning. When a skill is acquired *de novo*, new control policies are created, rather than recalibrating existing ones (Krakauer et al., 2019).

In a typical *de novo* task, participants are required to learn arbitrary, and usually non-intuitive associations between their movements and the outcomes to achieve the task goals (Wilterson & Taylor, 2021; Mosier et al., 2005; Velazquez-Vargas & Taylor, 2023). Crucially, these tasks are known to involve brain regions associated with declarative processes such as the hippocampus (Wise & Murray, 2000) and the prefrontal cortex (Fermin et al., 2016), also crucial for context-dependent learning (Heald et al., 2023b). This important distinction could allow *de novo* skill learning to be sensitive to contextual cues that are efficient in domains such as episodic memory, unlike visuomotor adaptation tasks which are known to involve a significant cerebellar-dependent component (Taylor et al., 2010)

Meta-learning (Wang et al., 2018) can be a unifying computational framework that formalizes *de novo* skill learning. In meta-learning, an agent (human or artificial) has to learn a distribution of different but related tasks. For *de novo* skill learning, the different tasks can use the same motor

repertoire but may map each motor movement to a different outcome. Within this framework, a meta-learning agent can either learn a single abstract representation across all tasks that accommodates all motor mappings or learn separate representations for all the unique motor mappings seen across tasks, where each representation can be bound to a particular external contextual cue (Musslick & Cohen, 2021). Meta-learning models are becoming increasingly more relevant in cognitive modeling of behavioral phenomenon due to their ability to implement Bayes-optimal learning algorithms on tasks for which Bayesian inference is intractable(Binz et al., 2023).

In the present study, we designed a grid navigation task to dissociate these two hypotheses. One group of participants (the context group) performed a grid navigation task (Fermin et al., 2010; 2016; Velazquez-Vargas et al., 2023) where they moved a cursor from start to target locations in two "worlds" randomly interleaved over trials (Figure 1). Each grid-world was associated with a unique key-mapping and had distinctive contextual cues. We compared performance on this group with another group of participants (no-context group) that experienced the trial-changes in key-mappings but not in the external contextual cues –i.e., grid worlds.

To implement cognitive models for learning to navigate with different visuomotor mappings, we trained two recurrent-based meta-learning agents using the architecture from Wang et al. 2018, which is an LSTM agent trained with reinforcement learning using an Actor-Critic framework (Mnih et al., 2016). The first model (context LSTM) incorporated external contextual information into its input while the second one did not (no-context LSTM). We tested their performance in the task and how well they predicted human participants' data. In addition, we examined how their internal representations gave rise to different strategies to learn multiple key-mappings, giving us potential insights and hypotheses for the neural representations of the participants.

We showed that humans and the meta-learners performed better in the task when provided with contextual information. We also found that, in both conditions, there is individual variability as to whether an individual acts more like the context LSTM or the no context LSTM. Additionally, we found that when no contextual information is provided, the internal representations of the LSTM agent are highly correlated in space and time while using the different key-mappings. However, these representations are less correlated when the LSTM agent that has the external context input. This suggests that the former is learning separate representations for different motor mappings whereas the latter is binding different motor mapppings to the same representation. Finally, we showed that the capacity to learn multiple key-mappings is dependent not only on the presence of contextual cues, but also on the complexity (number of hidden LSTM units) of the internal representations of the model. Our model-based neural analyses and human behavior results give us potential insight on how the brain can effortlessly learn multiple visuomotor mappings.

## 2 METHODS

### 2.1 PARTICIPANTS

Thirty two participants from Princeton University (13 males and 19 females, mean age = 21.7, sd = 3.8) were recruited through the psychology subject pool. The study was approved by the Institutional Review Board and all subjects provided informed consent prior to doing the experiment.

### 2.2 TASK

The experimental task was programmed using HTML/CSS/Javascript and hosted on Google Firebase. Visual stimuli were displayed on a 60 Hz Dell monitor using a Dell OptiPlex 7050'a machine (Dell, Round Rock, Texas) running Windows 10 (Microsoft Co., Redmond, Washington). Participants' responses were recorded using a standard desktop keyboard. Before the start of the experiment, the instructions were displayed on the screen.

The goal of the task was to move a cursor from start to target locations in a $9 \times 9$ grid environment using the least possible number of moves (Fermin et al., 2010; 2016; Velazquez-Vargas et al., 2023; Bera et al., 2021). To do so, participants used three keyboard keys (J, F and K) that moved the cursor to arbitrary and unknown adjacent locations (key-mapping): left-up, left-down and right (Figure 1). The task consisted of 360 trials where the start-target locations were randomly sampled out of eight possible pairs. For each pair, the target was placed seven moves away from the starting location

of the cursor. If participants arrived at the target using the minimum number of moves (optimal arrival), they would observe a happy emoji face at the target location and hear a pleasant sound. If they arrived at the target but not using the minimum number of moves (arrival), they would observe a neutral face and hear a neutral sound. If they did not arrive at the target in less than 10 s, the trial was terminated and participants would observe a sad face and hear an unpleasant sound.

Importantly for this study, participants performed the tasks using two key-mappings, which corresponded to different arrangements of the same moving directions (see Figure 1), and which were randomly interleaved across trials. Participants were not informed that there would be more than one key-mapping to perform the task. In order to test the effectiveness of external contextual cues in separating the learning of the visuomotor mappings, we split participants into two experimental groups.

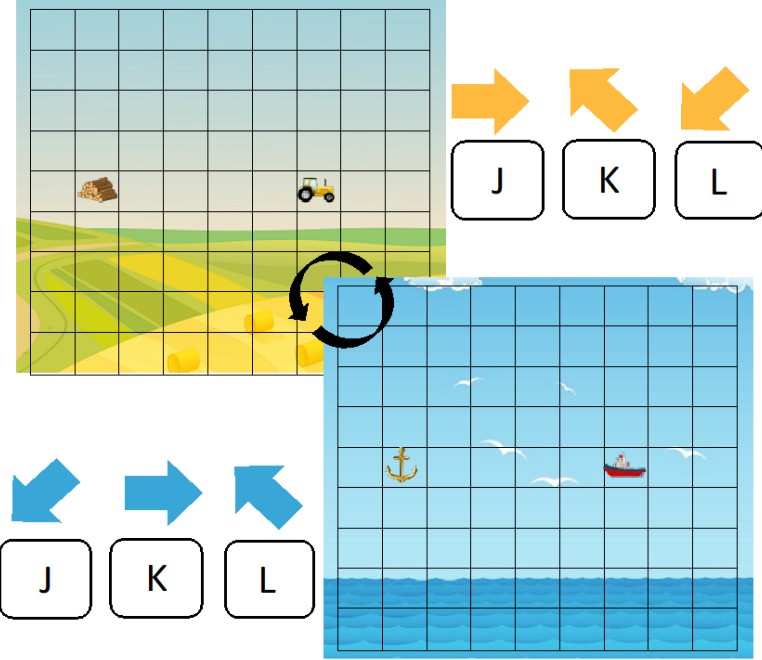

Figure 1: **Experimental task** Subject perform a grid navigation using different key mappings randomly interleaved over trials. In the context group, the key-mappings where deterministically signalled by a unique grid world, whereas in the no-context group, participants used both mappings in the same world.

**Context Group:** In this group (n =16) participants performed the grid navigation task in two different "worlds", either in the ocean or in a farm. Each world consisted of unique visual (background, cursor shape) and auditory (ambience song and cursor sound when moving) information. Most importantly, each world was deterministically associated with one of the key-mappings.

**No-context Group:** Participants in this group (n=16) performed the grid navigation task in a single world while still experiencing the changes in the key-mappings. At the beginning of the experiment, one of the worlds would be selected and remained throughout the whole task.

## 2.3 MODELS

Following previous work on meta reinforcement learning (Wang et al., 2018; Kumar et al., 2020), we used a Long short-term memory (LSTM) network trained using Advantage Actor Critic (A2C; Mnih et al. 2016. The recurrent structure of this agent, joint with a model-free learning mechanism, makes it suitable for sequential tasks like ours, where trial-and-error, reward-based processes are crucial to achieve the goals. The agents were implemented using Stable baselines 3 package (Raffin et al., 2019) with the Proximal Policy Optimization algorithm (PPO). We set constraints of the hyperparameters of PPO so that the algorithm becomes equivalent to A2C (Huang et al., 2022).

We used Optuna (Akiba et al., 2019) to tune the following hyperparameters: learning rate, discount factor gamma, GAE lambda, number of steps, batch size, entropy coefficient, the value function coefficient and the number of LSTM hidden units. After optimization, we trained the agents for $2 \times 10^5$ time steps, where they reached asymptotic performance. The reward structure was defined as follows: every move that did not lead to a terminal state (arrival to the goal) received $-1$, except when colliding with the grid walls, in which case the agent received a penalty of $-100$. If the agent reached the goal, it received a reward of $+10$.

We trained two types of agents that differ chiefly in the input provided to the LSTM network. The first one (context LSTM) received its current location and target locations on the grid as well as a context vector indicating which of the worlds was currently being observed. This agent faced the same situation as participants in the context group where the key-mapping changes were linked to changes in the contextual cues. On the other hand, the second agent (no context LSTM) only received its current location and the target location as input, mirroring the desing of the no-context group.

## 3 RESULTS

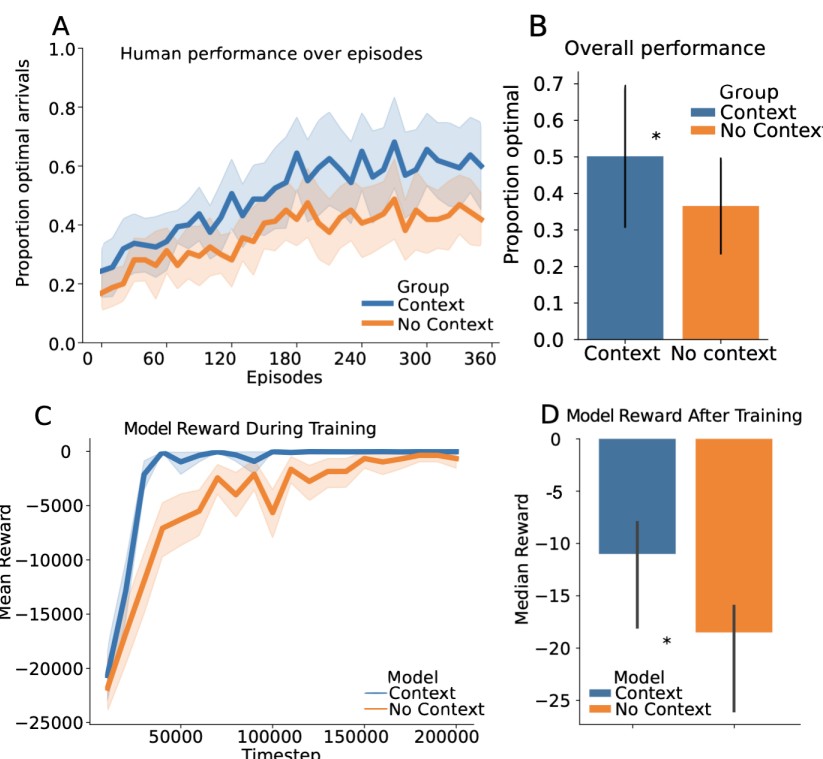

Figure 2: **Performance of Humans and Meta-RL Agents** (A). Mean performance of humans over the episodes they completed, measured by proportion of optimal arrivals. Shading represents 95% confidence intervals. Humans that were given context cues learned the task better. (B). Overall performance differences across different experimental groups in humans. Those in the context group did significantly better. (C). Mean reward over time during agent learning for both context and no-context agents. (D). Median reward after training. Agents that were given a external contextual cue for the mapping as additional input did significantly better.

### 3.1 HUMAN AND MODEL BEHAVIOR

As a metric of human performance in the context and no-context groups, we provide the proportion of times they arrive at the target optimally – i.e., using the minimum number of key presses– across

trials and in the entire task. Based on the optimal arrivals to the target, we observed that participants in both the context and no-context groups improved their performance over trials (Figure 2). However, the context group performed significantly better overall ($p = 0.03$). For the context LSTM and no-context LSTM agents, we provide the average reward per episode at different evaluation points across training, as well as at the end of training. The context LSTM and the no-context LSTM agents showed the same pattern as participants, where both models improved over the timesteps but the former performing overall better in the task ($p < 0.05$).

To identify the similarity between subjects' behavior and the context LSTM and no-context LSTM agents, we calculated the likelihood of subjects' responses according to the fully trained agents. That is, we performed a forward pass on the agents and obtained the likelihood that they would take the same action as the human. The likelihood for each action was averaged across all timesteps for each subject to produce one value indicating the average likelihood of a subject's response under the model. This analysis was done for both the context and the no-context LSTM agents.

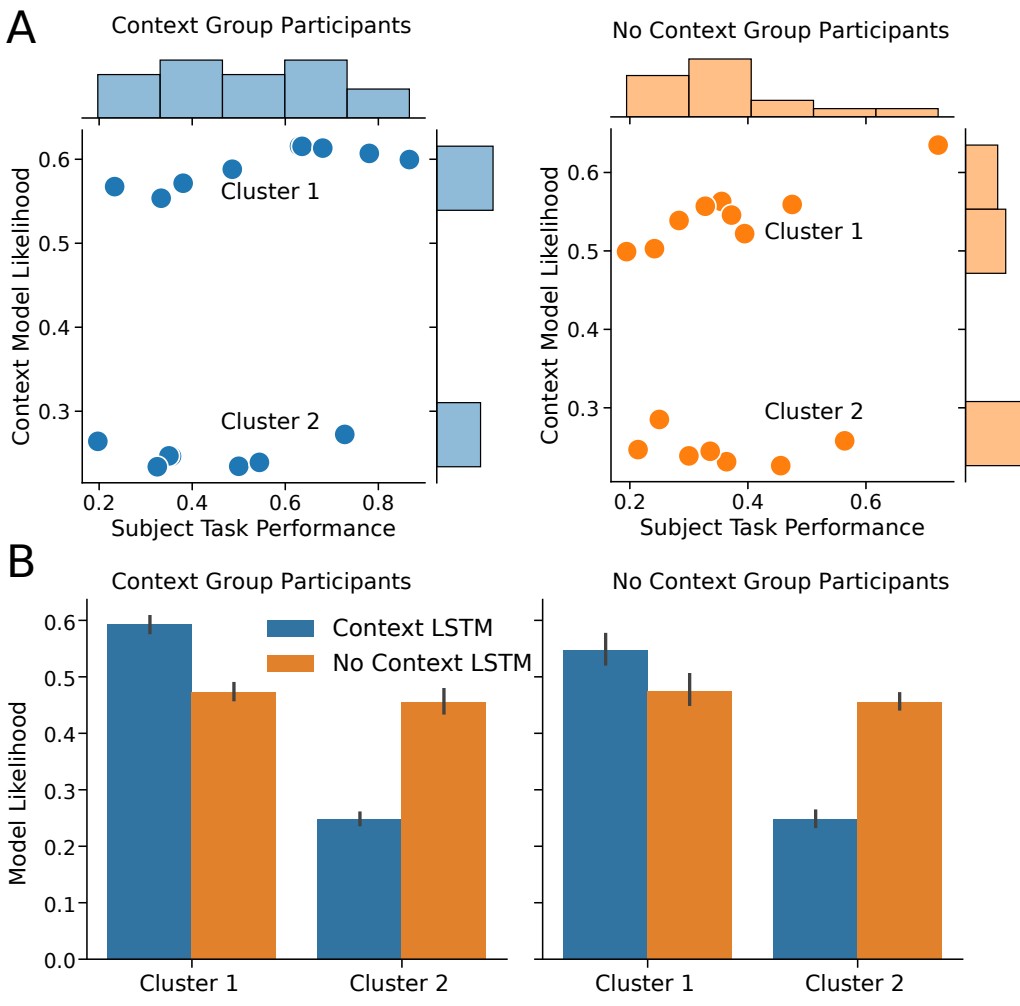

Figure 3: **In both experimental groups, humans are split between behaving like the context vs no context LSTM** (A). Joint scatterplot and histograms for likelihood of subjects' actions under the context LSTM model vs. performance of subjects on the task. In both experimental groups (whether participants received external context input or not), there are two clusters of participants — those whose actions are explained well by the context LSTM model and those whose are not. (B). If we examine the mean likelihood under both types of models (context vs no context LSTMs), we see that participants whose actions aren't as well explained by the context LSTM model have significantly higher likelihood under the no context LSTM model.

Figure 3A shows the average response likelihood for both experimental conditions under the context LSTM agent, where two clear clusters were formed. In Cluster 1, the context-LSTM agent was considerably better in predicting participants' responses than in Cluster 2, where it performed around chance level. Following this analysis, we found that participants that were poorly predicted by the context LSTM (Cluster 2) agent were instead significantly better predicted by the no-context LSTM ($p < 0.05$; Figure 3B). Likewise, for participants in Cluster 1, the context LSTM agent better predicted participants' responses compared to the no-context LSTM agent.

These results indicate that regardless of the experimental group, some participants where better predicted by the context model, suggesting that even in the absence of contextual cues from the experiment, participants could have relied on alternative cues to separate the representation under the different key-mappings. A point we we will address in the Discussion section.

## 3.2 MODEL REPRESENTATIONAL SIMILARITY ANALYSIS

We employed multiple analyses (Figure 4) to examine differences in representations between the cue and no-cue agent. Specifically, we first investigated how each network represents the same trial and environment under different motor contexts. To do this, we performed a Representational Similarity Analysis (RSA; Kriegeskorte et al. 2008 on both a spatial and temporal scale.

For the spatial RSA (Figure 4A), we evaluated the agents in $10^4$ episodes and computed the average hidden state (LSTM units) at each location in the $9 \times 9$ grid, separating the hidden states according to the key-mapping being used. For every grid state, we obtained the correlation coefficient between the averaged hidden states when using key-mapping A and the averaged hidden states when using key-mapping B. Lower correlations would indicate that agents represent the grid space between different action mappings more distinctively. In the context LSTM agent, the correlations were low across most grid states, with many locations on the grid showing no correlation, suggesting that spatial location is represented differently when using mapping A and mapping B. In contrast, for the no-context LSTM agent, the RSA generally depicts grid states with correlations that are significantly higher ($p < 0.001$), when the different key-mappings are being used compared to the context LSTM agent. The difference between the two suggests that the context LSTM agent has a distinct representation for grid states that depends on its current context, while the no-context LSTM model does not.

The temporal RSA (Figure 4B) provides complementary results showing that context is represented differently between models not just in space, but also in time. Specifically, we explored the similarity of hidden states of the agents as they progressed towards the target. To do so, we evaluated the agents in $10^4$ episodes and for every timestep we correlated the averaged hidden states when using key-mapping A with the averaged hidden states when using key-mapping B. In the no context LSTM agent, the representations on the initial timestep are highly correlated. As the agent steps towards the goal, representations become slightly less correlated, but for the most part remain highly correlated across the episode. In the context LSTM agent, however, representations under the different key-mappings drastically change with each step in the episode. Representations begin similar on the first time step (although less so compared to the no-context LSTM agent), but quickly drop to no correlation at around the 6th step in the episode. This suggests that the agent representing context increasingly leverages its context representation as it nears the goal state.

Taken together, these results show two different representational strategies, one where the agent attempts to learn distinct representations for each context, and another where the agent learns the task without an explicit representation of context.

## 3.3 MODEL REPRESENTATIONAL CAPACITY ANALYSIS

Given the capacity limitations in memory, computational resources and time, we would expect that humans exhibit constraints in the number of visuomotor mappings they can learn. To explore this idea, we investigated the agents' capacity to learn multiple key-mappings by varying the complexity of their internal structure, i.e. number of hidden units in the LSTM. In order to do so we exposed both the context LSTM and no-context LSTM agents to up to ten different key-mappings and using the following number of hidden units: $2, 5, 10, 20, 40, 80, 100, 120, 256$ or $512$. Each agent was

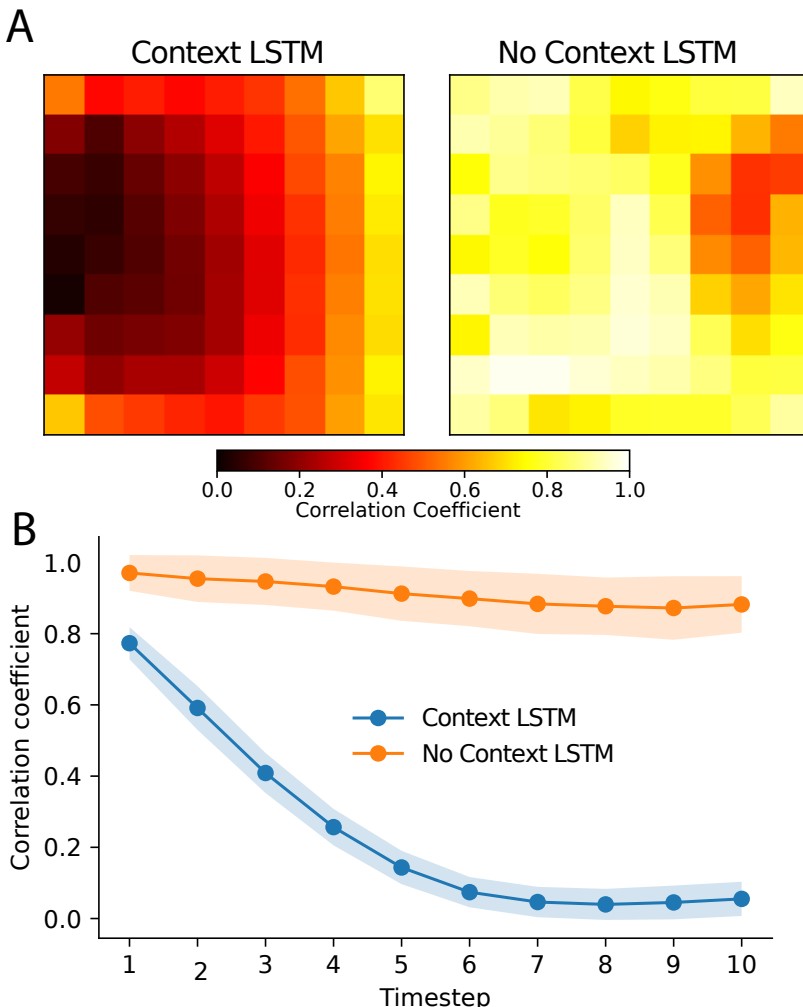

Figure 4: **Context agents have lower representational similarity under the different key-mappings over space and time** (A). Spatial RSA analysis. We correlated LSTM hidden state representations of the same episodes under the different key-mappings and showed the mean correlation across different spatial locations in the $9 \times 9$ grid. For context agents, this correlation is much lower, presumably because the representations of different key-mappings are more separated. (B). Temporal RSA analysis. We show how the correlation changes over time by plotting, for each time-point, the mean correlation of LSTM hidden state representations under the different key-mappings. For context agents, this correlation goes down over time whereas for no-context agents, there is less change overtime. This suggests context agents exhibit more dissimilar representational similarity under different key-mappings overtime.

trained for 200k timesteps followed by 100 evaluation episodes from which we report the averaged (normalized) reward.

Figure 5 displays performance of both models as the number of contexts and representational capacity increases. Across both models, we see better performance with more representational capacity. Between models, the model that explicitly receives contextual cues performs better as contexts increase than does the model that does not receive a cue. This is perhaps due to the way in which context is represented between both models. In the context model, each context is distinctly represented within the hidden layer, allowing for a unique task representation for each context. In contrast, the no context model does not distinctly represent each context. This supports work showing that unique task representations are necessary as the environment becomes more complex (i.e.

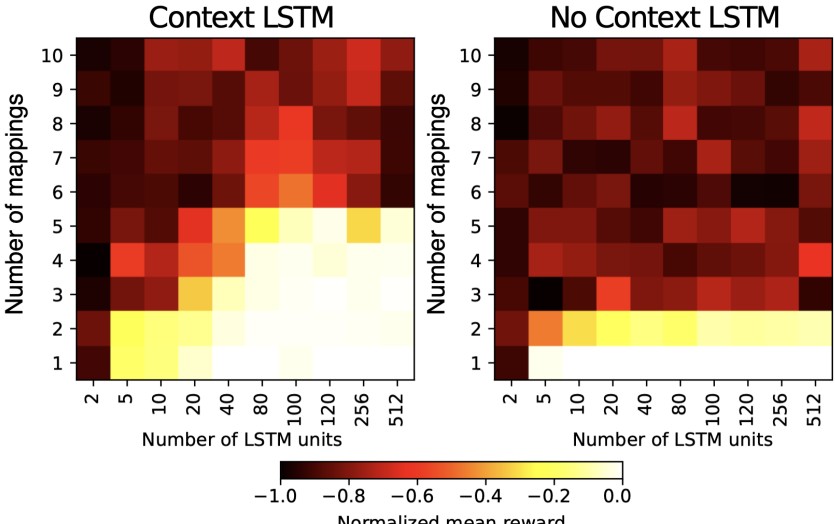

Figure 5: **For context agents, scaling the LSTM capacity enables learning more mappings** For both context and non-context agents, we varied the number of hidden units and exposed them to different number of mappings during training. For context agents, increasing the number of LSTM units allows for learning of more mappings until about 5. For no-context agents, varying the number LSTM units does not have as strong an effect.

when the number of contexts increase) and adds that explicit cues aid in both the speed at which the task is learned and how it is represented.

## 4 DISCUSSION

Humans posses a remarkable capacity to learn complex, and often arbitrary, visuomotor mappings. For example, when learning to play video games humans are able link the button presses from the controller with arbitrary actions in the game. What is more, not only are they capable of learning such arbitrary mappings and use them to achieve goals, but they can build an entire repertoire of them. It is not uncommon to witness players seamlessly transitioning between mappings for distinct gaming genres, be it from a first-person shooter to a racing or football game.

In the current work, we have shown that the acquisition of distinct visuomotor associations can be aided by external contextual cues. In particular, participants' that are provided with contextual information that deferentially cues the key-mappings, performed significantly better than participants without it. These results are supported by previous findings outside the motor domain where external contextual cues allow subjects to discriminate between different contingencies of the task (Gershman, 2017; Bornstein & Norman, 2017). Similar cues, however, do not prevent catastrophic interference in visuomotor adaptation (Howard et al., 2013), except under specific circumstances (Avraham et al., 2022). We believe this effect arises due to the fundamental distinction between adaptation and *de novo* learning, where the former recalibrates an existing control policy and the second one builds one from scratch. This can generate qualitative differences between the contextual cues the systems supporting each process are sensitive to. On the one hand, cerebellar dependent recalibration processes could be more easily influenced by contexts related to the body states or the world kinematics Howard et al. (2013). However, *de novo* learning, particularly in the domain of spatial navigation likely involving the hippocampus and the prefrontal cortex (Fermin et al., 2016), could be more sensitive to visuospatial contextual cues.

It is important to emphasize that while the context group outperformed the no-context group, the latter still improved its performance over trials, arriving optimally a greater proportion of times toward the end of the experiment ($p < 0.05$). This improvement occurred in the absence of external contextual cues, which could aid in differentiating the key-mappings. We believe this result could be attributed to participant relying on a different set of cues: movement-related ones. In particular,

given the specific movement directions that we used for the key-mappings (left-up, left-down and back), it is possible to arrive at the target locations with multiple and equivalently optimal trajectories which begin with a different action. For example, for a target that is right above the cursor, pressing the key that moves right, or left-up, would leave the cursor one move away from the target. Although this move makes no difference in terms of performance, it does provide information about the ongoing mapping, given that no key moves to the cursor to the same direction. Therefore, participants could have used the first move of the cursor to sample the ongoing key-mapping and adapt the subsequent moves accordingly. In effect, this strategy would have provided participants with contextual information similar to the context group and would have been able to learn the two mappings separately. This is consistent with the fact that some participants in the no-context group had a higher likelihood under the context LSTM than the no context LSTM (Figure 3A). Alternatively, participants could have learned a single of the two key-mappings, or an averaged version of them, which over time would have allowed them to improve in the task, although not to the same extent as the context group. Participants relying on the latter strategies would be learning, in effect, a single key-mapping. Further analysis would be needed to test these hypotheses.

In order to have a behavioral model for the presence or absence of contextual cues while acquiring multiple mappings, while also providing insight into the neural representations, we leveraged recent developments on meta-reinforcement learning (Wang et al., 2018; Huang et al., 2022; Kumar et al., 2020). Mirroring human performance, we found that an LSTM agent that explicitly receives information about its context outperforms an LSTM agent that does not. Moreover, we found that participants that were poorly described by the context LSTM, either in the context or no-context groups, were instead best described by an LSTM model with no contextual information. Through implementing these models, we were able to find that within both experimental groups, people vary in either producing behavior consistent with a model learning separate vs shared representations across different mappings (Fig. 3).

Based on the spatial and temporal RSA, we would expect to see differences within neural representations of participants relying on contextual information (either from the experiment or potentially from their own actions) vs. individuals that don't. However, due to the fact that the no context LSTM explained some participants' behavior better than the context LSTM (Figure 3), it is conceivable that some participants held a single representation in the presence of contextual cues, while others held different representations in the absence of them. This may beg the question — why would a participant choose to hold a single representation even when given contextual cues? The fact that the number of mappings learned scales with the capacity of the model (Figure 5) suggests that learning these separate representations uses more cognitive resources. Subjects choosing to not learn separate representations may be behaving rationally according to their cognitive resources (Lieder & Griffiths, 2020). From our capacity analysis of the networks in Figure 5, we predict that participants who don't learn separate representations may take longer to learn the same number of mappings than participants who take advantage of the contextual cues to learn separate representations. Confirming these predictions with brain imaging, and generally studying how participants share vs. separate neural representations of visuomotor mappings to manage time and computational resources, will be a rich line of future work.

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
