# OpenReview forum: "Learning to Abstract Visuomotor Mappings using Meta-Reinforcement Learning"
_ICLR.cc/2024/Workshop/Re-Align — ICLR 2024 Workshop Re-Align Poster_

### Official Review · Reviewer_ELTb · 2024-02-20

**Rating:** 2
**Fit:** 3
**Confidence:** 2

**Workshop Review:**

Summary
The paper attempts to model the learning of visuomotor mappings with/without contextual cues in humans and artificial RL agents.
By training humans and meta-learning agents on grid-world navigation tasks with different key mapping cued by context, it shows contextual information helps better learning and different representational strategies of artificial agents in presence of contextual cues.

Strong points:
The motivation is well defined and interesting; distinguishing de-novo skill learning vs. visuomotor adaptation.

Weak points:
Shallow analysis of representational difference acquired by RL agents.
The figures can be more descriptive (missing axes, units etc).

I will accept the paper as it is well-aligned with the workshop scope and the preliminary results are interesting, but in-depth analysis of representational strategy than simple RSA will make the paper stronger; if it's different, how is it different? Further analysis on effective dimensionality /geometry might give more insights. Section 3.3 on representational capacity seems to be trivial and it is not clear what this section delivers to the scope of the paper.

**Reason For Not Giving Higher Score:**

N/A

**Reason For Not Giving Lower Score:**

N/A

**Reviewer Domain:**

neuroscience

---

### Official Review · Reviewer_5Xqv · 2024-02-23
**Interesting approach**

**Rating:** 2
**Fit:** 3
**Confidence:** 2

**Workshop Review:**

The authors conduct an grid-navigation experiment where an LSTM-based agent, and humans, interact with the environment through three actions that have an effect in the environment that is not specified to the participant. There are two possible mapping forms action to translation in the navigation environment - the sets are sampled randomly, and for one group of agents/participant, this is coupled to a change in contextual information (change in background); for the other group it is not.
The authors investigate whether or not contextual information a) linked to improved performance b) predictive of de-novo skill learning vs adaptation. The authors demonstrate the latter by showing the change in representational similarity between the LSTM hidden state in tasks with different action key-mappings. The authors also show LSTM hidden state size is correlated with the capacity to perform better when the number of contexts is increased, if context is given to the agent.

I
Strengths: The paper is well written, and I think looking at meta learning as a paradigm for studying skill acquisition is well-grounded.

Weaknesses: The authors claim: “Our model-based neural analyses and human behavior results give us potential insight on how the brain can effortlessly learn multiple visuomotor mappings”, which is not substantiated by the experiments.

There also needs to be a deeper analysis of the strategy of the LSTMs and participants with respect to the fact that sampling one action is all the is required to determine the key mapping. The authors correctly identify this, but I think this is essential for publication.

Questions: have the authors considered  using CCA over RSA for comparing LSTMs? https://arxiv.org/abs/1806.05759
What the state coverage of the agents? I.e. when comparing activations in the grid
When averaging over space, how do the authors account for the fact  that LSTM hidden state will depend on the previous states seen?

**Reason For Not Giving Higher Score:**

I believe this paper does not fully substantiate its hypothesis

**Reason For Not Giving Lower Score:**

The paper should still be included in the workshop as the technique and approach will be interesting to the audience

**Reviewer Domain:**

machine learning

---

### Decision · Program_Chairs · 2024-03-02

Accept (Poster)